# Molecular Characterisation and Phylogeny of Tula Virus in Kazakhstan

**DOI:** 10.3390/v14061258

**Published:** 2022-06-09

**Authors:** Nur Tukhanova, Anna Shin, Nurkeldi Turebekov, Talgat Nurmakhanov, Karlygash Abdiyeva, Alexandr Shevtsov, Toktasyn Yerubaev, Gulnara Tokmurziyeva, Almas Berdibekov, Vitaliy Sutyagin, Nurbek Maikanov, Andrei Zakharov, Ilmars Lezdinsh, Lyazzat Yeraliyeva, Guenter Froeschl, Michael Hoelscher, Stefan Frey, Edith Wagner, Lukas Peintner, Sandra Essbauer

**Affiliations:** 1Center for International Health, University Hospital, Ludwig Maximilians University, 80336 Munich, Germany; tukhanovanur@gmail.com (N.T.); annashin86@gmail.com (A.S.); guenter.froeschl@med.uni-muenchen.de (G.F.); 2Aikimbayev’s National Scientific Center for Especially Dangerous Infections, Almaty 050054, Kazakhstan; nurik_1976@mail.ru (N.T.); nti0872@gmail.com (T.N.); gdirect@nscedi.kz (T.Y.); zgd-1@nscedi.kz (G.T.); 3Almaty Branch National Center for Biotechnology, Almaty 050054, Kazakhstan; karla.abdi@yandex.kz; 4National Center for Biotechnology, Nur Sultan 010000, Kazakhstan; ncbshevtsov@gmail.com; 5Taldykorgan Antiplague Station, Branch Aikimbayev’s National Scientific Center for Especially Dangerous Infections, Taldykorgan 040000, Kazakhstan; tpcstald@mail.ru (A.B.); vit197803@mail.ru (V.S.); tpcstald.12-zoootdel@mail.ru (I.L.); 6Oral Antiplague Station, Branch Aikimbayev’s National Scientific Center for Especially Dangerous Infections, Oral 09002, Kazakhstan; nmaikanov@mail.ru (N.M.); awzacharow@mail.ru (A.Z.); 7National Scientific Center of Phthisiopulmonology, Almaty 050010, Kazakhstan; l.eralieva@mail.ru; 8Division of Infectious Diseases and Tropical Medicine, University Hospital, Ludwig Maximilians University, 80802 Munich, Germany; hoelscher@lrz.uni-muenchen.de; 9Bundeswehr Research Institute for Protective Technologies and CBRN Protection, 29633 Munster, Germany; stefan1frey@bundeswehr.org; 10Section of Experimental Virology, Institute of Medical Microbiology, Jena University Hospital, 07745 Jena, Germany; edithwagner@bundeswehr.org; 11German Centre for Infection Research, Department Virology and Intracellular Agents, Munich Partner Site Bundeswehr Institute of Microbiology, 80937 Munich, Germany; sandraessbauer@bundeswehr.org

**Keywords:** orthohantavirus, rodents, Republic of Kazakhstan, Tula virus

## Abstract

Orthohantaviruses are zoonotic pathogens that play a significant role in public health. These viruses can cause haemorrhagic fever with renal syndrome in Eurasia. In the Republic of Kazakhstan, the first human cases were registered in the year 2000 in the West Kazakhstan region. Small mammals can be reservoirs of orthohantaviruses. Previous studies showed orthohantavirus antigens in wild-living small mammals in four districts of West Kazakhstan. Clinical studies suggested that there might be further regions with human orthohantavirus infections in Kazakhstan, but genetic data of orthohantaviruses in natural foci are limited. The aim of this study was to investigate small mammals for the presence of orthohantaviruses by molecular biological methods and to provide a phylogenetic characterization of the circulating strains in Kazakhstan. Small mammals were trapped at 19 sites in West Kazakhstan, four in Almaty region and at seven sites around Almaty city during all seasons of 2018 and 2019. Lung tissues of small mammals were homogenized and RNA was extracted. Orthohantavirus RT-PCR assays were applied for detection of partial S and L segment sequences. Results were compared to published fragments. In total, 621 small mammals from 11 species were analysed. Among the collected small mammals, 2.4% tested positive for orthohantavirus RNA, one sample from West Kazakhstan and 14 samples from Almaty region. None of the rodents caught in Almaty city were infected. Sequencing parts of the small (S) and large (L) segments specified Tula virus (TULV) in these two regions. Our data show that geographical distribution of TULV is more extended as previously thought. The detected sequences were found to be split in two distinct genetic clusters of TULV in West Kazakhstan and Almaty region. TULV was detected in the common vole (*Microtus arvalis*) and for the first time in two individuals of the forest dormouse (*Dryomys nitedula*), interpreted as a spill-over infection in Kazakhstan.

## 1. Introduction

The genus *Orthohantavirus* (family *Hantaviridae*, order *Bunyavirales*) includes zoonotic pathogens. This group of viruses plays an important role in causing human diseases worldwide. Orthohantaviruses are single-stranded negative polarity RNA viruses, and the genome consists of three segments. The large (L) segment encodes a viral RNA-dependent RNA polymerase, the medium (M) segment encodes the glycoprotein precursor (GPC), which is processed to the glycoproteins Gn and Gc, and the small (S) segment encodes the nucleocapsid (N) protein [1].

Small mammal species are a reservoir for orthohantaviruses. Orthohantaviruses are presently known to infect rodents (subfamilies Murinae, Arvicolinae, Sigmodontinae, and Neotominae), but are also detected in different shrews and moles [2,3,4]. In Eurasia, humans are infected either by rare direct contact or indirectly by inhalation of orthohantaviruses containing dust from dried excreta [5,6].

Old World orthohantaviruses can cause haemorrhagic fever with renal syndrome (HFRS) and are mainly transmitted by members of the Murinae and Arvicolinae subfamilies [2,3]. In Europe, the main causative agent of HFRS is Puumala virus (PUUV) causing nephropathia epidemica (NE), a mild form of HFRS. A mild to severe form of HFRS is caused by Dobrava-Belgrade virus (DOBV). In Asia, the most relevant species is Hantaan virus (HNTV) that causes a severe form of HFRS. Seoul virus (SEOV) is distributed worldwide and can cause a moderate form of HFRS [6,7,8,9]. Pathogenicity of Tula virus (TULV) to humans is limited, only few reports of human cases were described in Europe [10,11,12,13], despite the fact that TULV is found in Asia and Europe. In North America, the TULV-related Prospect Hill virus was identified in a *Microtus* species (*M. pennsylvanicus*) but no human infections have been reported here either [2,7,14,15].

The Central Asian Republic of Kazakhstan has a vast territory and contains several types of landscapes such as forest-steppes, steppes, semi-deserts, deserts, and mountain ranges [16,17]. In these different geographic settings, Kazakhstan has numerous natural foci of important zoonotic pathogens such as *Yersinia pestis, Bacillus anthracis, Francisella tularensis, Leptospira, Listeria monocytogenes*, tick-borne encephalitis virus (TBEV), Crimean-Congo haemorrhagic fever virus (CCHFV), and orthohantaviruses [17,18].

An investigation of small mammals on the Dzungarian Alatau mountain range in Almaty region in 1990–1993 showed that some rodents contain orthohantavirus antigens (*n* = 644, 5.3%) [19]. Twenty years later, a study conducted in the same region using antigen assays found traces of orthohantavirus antigens in 2.2% of investigated tissue suspensions of rodents collected in 2010–2016 [20,21]. Furthermore, the existence of Tula virus was proven in tissue samples of *Microtus arvalis* in Almaty region (periphery of Taldykorgan city and Karatal village) [22].

The first human case of HFRS was detected in the year 2000 in the Zharsuat village in the Borili district, a part of the West Kazakhstan region [23,24]. Further investigations of host reservoirs were started, and from 2001 to 2011 almost 50,000 small mammals including 30 species were screened for the presence of orthohantavirus antigen. A total of 1.53% of different species, mostly *Myodes glareolus*, *Microtus arvalis, Microtus minutus*, *Apodemus uralensis*, and *Mus musculus* were positive. Therefore, so far, natural foci of orthohantaviruses were described in the four northern districts of the West Kazakhstan region (Borili, Bayterek, Shyngyrlau, and Terekti) and very preliminary in the Aktobe region [25,26]. However, in all investigations on orthohantaviruses in West Kazakhstan, contemporary molecular methods were never applied.

To date, there have been no officially registered human cases of HFRS in the Almaty region. However, an investigation of patients with fever of unknown origin (FUO) in Almaty and Kyzylorda regions showed orthohantavirus-reactive antibodies in sera of patients. This indicates that orthohantaviruses might also be endemic in the southeast of Kazakhstan [23].

The aim of this study was to investigate small mammals for the presence of orthohantaviruses by molecular biological methods in the Almaty region, including Almaty city and in West Kazakhstan, representing an officially endemic region for orthohantavirus infections in humans.

## 2. Materials and Methods

### 2.1. Study Setting and Rodent Sampling

Small mammals were trapped in 2018 and 2019 in West Kazakhstan (Bayterek, Borili, Terekti, and Taskaly districts: 19 trapping sites), Almaty region (surroundings of Tekeli city, Rudniychniy, and Bakanas: four trapping sites) and Almaty city (seven trapping sites) during spring, summer, autumn, and winter seasons (Figure 1 and Appendix A).

Snap traps were set overnight at 5 m intervals baited with cured pork fat. In the early morning, captured small mammals were collected, stored on dry ice, and transported to the laboratory for immediate processing. After morphological identification of the species, necropsy was performed, and internal organs (lung, heart, brain, kidney, liver, spleen, ears, and transudate) were aseptically collected and stored in RNA later (Thermo Scientific, Langenselbold, Germany) at −20 °C until further use [27].

### 2.2. RNA Extraction, PCR Amplification and Sequencing

Lung tissue samples were homogenized in 1 mL MEM for 2 min at 30 Hz in a TissueLyser II (Qiagen, Hilden, Germany). RNA was extracted from 140 µL homogenized supernatant using a commercial QiAmp Viral RNA Mini Kit (Qiagen, Hilden, Germany) according to manufacturer’s instructions. To determine the sequences of parts of the S and L segments, RNA was reverse-transcribed and amplified using primers detecting a variety of orthohantaviruses and subsequently sequenced using terminator cycle sequencing. In detail, for the S segment, a conventional PCR was applied using Superscript III one step RT-PCR system with Platinum Taq high fidelity polymerase (Invitrogen, Langenselbold, Germany) and the primers DOBV-M6 (5′-AGYCCWGTNATGRGWGTRATTGG-3′) and DOBV-M8 (5′-GAKGCCATRATNGTRTTYCKCATRTCCTG-3′), as described elsewhere [28,29]. The RT-PCR products were analysed using a 1.5% agarose gel with an expected amplicon size of 380 base pairs (bp). To detect a partial L-segment sequence (230 bp), a real-time RT-PCR using a Qiagen One Step RT-PCR mix was performed. Here, the primer-mix contained forward (1a-fw: 5′-TGATGCATATTGTGTGCAGAC-3′, 1b-fw: 5′-TGATGCATACTGTGTGCAAAC-3′, 1c-fw: 5′-CAGTATGATGCATACTGTGTCCAA-3′, 1d-fw: 5′-TGATGCCTATTGTGTTCAGAC-3′) and reverse (1a-rev: 5′-CTTGCTCTGTTTTGAATCTCA-3′, 1b-rev: 5′-CTTGCTCGGTGTTGAATCGCA-3′, 1c-rev: 5′-CCTGTTCTGTATTAAATCTCA-3′, 1d-rev: 5′-CTTGTTCAGTCTTGAATCTCA-3′) (0.125 µM each) primers, complemented with EvaGreen (VWR International, Vienna, Austria) as PCR reagents [30].

To confirm the species determination of the small mammals, a *cytochrome b* (mt-Cytb) gene sequencing was applied as described in [31]. For analysis of the mitochondrially encoded Cytb, supernatant from homogenised rodent lung tissue in elution buffer (Qiagen, Hilden, Germany) was used. A total of 400 ng of DNA were amplified by PCR using the primer combination Cytb-Uni-fw (5′-TCATCMTGATGAAAYTTYGG-3′) and Cytb-Uni-rev (5′-ACTGGYTGDCCBCCRATTCA-3′) targeting an approximately 1000 bp long fragment. The PCR was enabled by using the Invitrogen Platinum Taq High Fidelity DNA Polymerase (ThermoFisher Scientific, Langenselbold, Germany).

All positive PCR products (fragments of the S and L segment, Cytb fragments) were purified using a QIAquick PCR purification Kit (Qiagen, Hilden, Germany) and sequenced according to the manufacturer’s instructions by using a BigDye Terminator v3.1 Cycle Sequencing Kit (Applied Biosystems, Langenselbold, Germany) and a 3730xl DNA Analyzer (Applied Biosystems, Langenselbold, Germany).

### 2.3. Phylogenetic Analysis

The generated nucleotide sequences were aligned using the ClustalW method in Bioedit 7.2.5. Prior to alignment, the sequences were trimmed for the primers resulting in final sequence lengths of 346 nucleotides (nt) for the S segment and 184 nt for the L segment that were then used for the phylogenetic analysis. Phylogenetic trees were constructed in MEGA X with the Maximum Likelihood method based on the Tamura 3-parameter model [32]. These analyses involved published S and L segment nucleotide sequences from GenBank trimmed to the same length with accession numbers listed in the captions to Figure 2 and Figure 3. To set an outgroup in the phylogenetic trees, sequences of PUUV S and L segments, also trimmed to the respective lengths, were used (NC005224 and NC005225, respectively).

## 3. Results

In total, 621 small mammals were collected in nine sampling areas, at all together 30 trapping sites during the years of 2018–2019 (Table 1).

These small mammals represent eleven species from four families: Cricetidae (*M. arvalis*, *M. glareolus*, *M. kirgisorum*), Muridae (*A. uralensis*, *M. musculus*, *R. norvegicus*, *M. meridianus*), Gliridae (*D. nitedula*) and Soricidae (*S. araneus*, *S. minutus*, *C. suaveolens*). Sex distribution of collected mammals was almost equal with 59% male and 41% female.

Out of all 621 collected small mammals 15 (2.4%) were positive for orthohantavirus RNA (Appendix A). In Almaty city, all analysed rodents failed to yield a positive result. The infected individuals represented two species, *M. arvalis* (*n* = 13, 15.1%) and *D. nitedula* (*n* = 2, 13.3%) (Table 2). Three *M. arvalis* and both of the orthohantavirus carrying *D. nitedula* samples were further tested by *cytochrome b* gene-specific PCR and subsequent sequence analysis [31] to confirm the morphological determination. The Cytb sequence of Tekeli23 *M. arvalis* (ON513439) was 99% similar to a nucleotide sequence of *M. arvalis* originating from Russia, Ekaterinburg (MG703092). Both the *D. nitedula* Tekeli17 (ON513437) and Tekeli20 (ON513438) species were also confirmed by mitochondrial *cytochrome b* sequencing. The two sequences are 98% identical to a sequence from *D. nitedula* described from Mongolia (LR131101). All orthohantavirus infected specimens where either adults (*n* = 11) or sub-adults (*n* = 4).

A partial S segment sequence analysis revealed that all 15 small mammals harboured RNA of TULV. The obtained sequences were aligned with published TULV partial S segments available for Central Asia, Eastern and Central Europe, and China. These included clades from different geographic regions such as Central North (CEN.N), Eastern North (EST.N), Central South (CEN.S), Eastern South (EST.S), Eastern Carpathian, Russia (Tula, Crimea, Samara, and Omsk), Lithuania, and China (Xinjiang) (Figure 2). A nucleotide sequence identity matrix of the detected S segments compared with sequences of geographically relevant regions reveals that the sequences have an identity range from 78.9–100% (Table 3).

By comparing the newly identified TULV sequences with published genomes, four clusters can be classified that are geographically relevant for Kazakhstan (Figure 2): (I) The South-East Kazakhstan cluster consists of new virus sequences from Tekeli and Rudnichniy and already published sequences from Taldykorgan (AM945879) and Karatal (AM945877, AM945878) with a nucleotide sequence identity range of 94.3–100%. (II) The second neighbouring cluster from China and Russia includes sequences from Xinjiang (KX270414, MN052670) and from Omsk in Russian Siberia (AF442621) with a nucleotide sequence identity ranging from 84.5–87.5% within the cluster. (III) The third cluster are sequences from the Tula area of Russia (Z30941-4) and from Crimea (KJ742928) with an identity range of 87.5–98.5%. (IV) One positive sample (*M. arvalis*, Bayterek-56 07/19) from West Kazakhstan had a 93.4% sequence identity with the Samara virus from Russia (DQ061258). These two virus sequences form a separate cluster from all the other sequenced viruses (Figure 2).

A 78.9–99.4% nucleotide sequence identity is noticeable between the cluster of southeast Kazakhstan (I) that contains genomes form China and Siberia (II), as well as among the clade of Tula and Crimea area of Russia (III) and with the new sequence from West Kazakhstan (IV). The sequences from southeast Kazakhstan (I) are 75.8–99.1% similar to the Samara virus of Russia (IV).

The sequences from West Kazakhstan have 84.5–98.5% identity with variants from the Tula region and Crimea (III) and 82.1–87.5% identity with genomes from China and Siberia (II), respectively.

In silico translated S segment sequences of all TULV sequences included in this study showed 86–100% amino acid sequence identity for the N protein to other variants (Appendix A).

Similarly, the sequences of parts of the L segment from Almaty and West Kazakhstan regions were aligned with other L segment sequences available from GenBank. These resulted in four clusters of TULV from various geographic locations. Sequences of the 14 samples from Almaty region grouped in one subcluster (South-East Kazakhstan, I), sequences from China (Xinjiang, MN183133-6) and Turkey (Palandoken, MH649272) in a second cluster (II). These sequences show nucleotide sequence similarities of 80–99.3%. One sample from West Kazakhstan (Bayterek-56 07/19, *M. arvalis*, III) grouped distant from the other sequences (Figure 3) and had a nucleotide sequence similarity of 80.6–99.3% to the samples from South-East Kazakhstan (I) (Table 4).

By translating these nucleotide sequences into its short peptide sequence of 61 amino acids, two recurring substitutions become apparent. The sequences Tekeli-110 (OL677529) and Rudnichniy-94 (OL677532) show at position 1760 a P versus R exchange and at position 1773 a K versus E aberration in comparison to published consensus sequences (Appendix A).

## 4. Discussion

We designed a study to screen for orthohantavirus RNA in small mammals in the Republic of Kazakhstan regions West Kazakhstan, Almaty region, and Almaty city. Here, we demonstrate for the first time the presence of TULV in West Kazakhstan and confirm it in the Almaty region in Kazakhstan. The rate of positive individuals of *M. arvalis* is 15.1% (13/86), which agrees with previous studies [33,34]. Among all positive samples, males accounted for 60% (*n* = 9), which is consistent with other studies showing that male small mammals have a greater infection rate for orthohantaviruses (Table 2) [35].

West Kazakhstan is the only official orthohantavirus endemic region with registered human cases of orthohantaviruses infections so far [36,37]. Long-term investigations of host reservoirs starting from 2001 by colleagues from the Oral antiplague station revealed natural foci of orthohantaviruses in the floodplains of the Ural River. This area directly borders the Russian Orenburg and Samara regions, where orthohantavirus is also endemic [26,38]. Several small mammals that are also spread in this region such as *M. glareolus*, *M. arvalis*, *A. uralensis*, and *M. musculus* contained orthohantavirus antigens [26]. Our study could confirm the existence of TULV in West Kazakhstan region in *M. arvalis*, but only in one specimen. Actually, we expected to find the presence of PUUV, due to clinical manifestations of hospitalized patients with HFRS that is primarily caused by PUUV. Additionally, *M. glareolus*, the main host reservoir of PUUV is very common in this region. However, the number of captured *M. glareolus* and other small mammals was rather low to draw a statistically convincing picture on the spread of orthohantavirus in this area. Still, this study is the first to perform molecular-biological methods in the region of West Kazakhstan and generated the first orthohantavirus sequence from TULV [26,37].

In this study, for the first time, small mammals were screened for the presence of orthohantaviruses in Almaty city, but no positive results were revealed in the captured rodent species that were *M. kirgisorum*, *A. uralensis*, *R. norvegicus*, and *M. musculus*. The latter where the most captured animals in Almaty in this study. All these species might carry different orthohantaviruses such as, e.g., SEOV, but the primers used in this study are detecting all species of orthohantaviruses as shown in an internal validation of the primer sets for certified diagnostics [39]. The reason why there were no traces of orthohantavirus detected in the city are manifold but may rest in the different living conditions and species composition of the rodent population. However, as PUUV-reactive antibodies were found in a retrospective study in patients with fever of unknown origin [23], further studies have to be conducted in different geographic areas of Almaty city in order to unveil the real prevalence in the city.

Nevertheless, in the Almaty region, an area stretching north of Almaty city, TULV was identified and sequenced in several specimens captured in Tekeli city and Rudnichniy village. All TULV RNA was detected in two different species of small mammals, *M. arvalis* and *D. nitedula*. *M. arvalis* is a commonly known host for TULV. Interestingly, however, we also found TULV in *D. nitedula* of the Gliridae family that represents a novel host species for TULV. A cytochrome b sequence analysis confirmed the species. So far, the literature only reports on TULV in species belonging to the Arvicolinae subfamily, such as *Microtus* spp. and *Lagurus lagurus* [40,41]. However, by comparing the capture sites of those two infected specimens, it becomes apparent that the spots in Tekeli had a spatial distance of only 325 m. In this region, *D. nitedula* is a common mammal, mostly living on trees but also reported to hunt for edibles on the ground, since also the traps were only located on terra firma. There, it may have indirect contact with *M. arvalis* that builds nests in subterraneous burrows but also gathers edibles on the ground. The infection of atypical host species with orthohantavirus is designated as a spill-over infection and is reported in high incidence areas in Europe [41]. Since we identified several infected rodents in the Tekeli area and the S segment sequences derived from *D. nitedula* and *M. arvalis* are almost identical, such a spill-over event is in the scope of possibilities [42,43,44]. Nevertheless, this result implies the need for a more extensive follow-up host-study for infected small mammals in the area of the Almaty region to obtain information on the actual distribution of orthohantaviruses in this area.

To further estimate the connection of these viruses, we performed sequencing of parts of the S and L segments. Sequence similarities for the partial S segments of the clusters of South-East Kazakhstan (I) and West Kazakhstan/Samara (IV) resemble these of previous studies [41]. Furthermore, the phylogenetic analysis of the partial S segment sequences enabled the classification of TULV in a broad geographical range [43,45,46]. Our results highlight that TULV from West Kazakhstan is indeed in close evolutionary relationship with TULV described in Samara, the adjoining region in the Russian Federation (DQ061258). Almaty region (Tekeli and Rudnichniy) has its own cluster separated from all other TULV sequences for the S segment (Figure 2). Additionally, it is evident that the West Kazakhstan TULV S segment sequence is only distantly related to other Kazakhstan sequences as, for instance, from the Almaty region, a region over 2000 km apart from West Kazakhstan. Sequences from the Tekeli city and Rudnichniy village in the Almaty region shared a close relationship with previously published sequences of *M. arvalis* sampled in the village of Karatal and Taldykorgan city, located also in the Almaty region [22]. It is highly probably that there exist different geographic lineages of TULV in Kazakhstan transmitted by different lineages of rodents as recently shown for TULV sequences in Europe [33,40,41].

The sequence relationships identified for the S sequence analysis can also be identified in the analysis of the partial L segment sequences, where we could show that the TULV L segment sequence from West Kazakhstan region formed its own distinct geographic cluster. In general, published sequences for the L segment in this region are sparse and for the Almaty region, we describe for the first time also TULV L segment sequences, in comparison to a previous study that only analysed the S segment [22]. Sequences from Tekeli and Rudnichniy in Almaty region cluster in an individual branch in one big cluster with sequences from China and Turkey (Figure 3) [47,48]. This finding goes along with previous studies who have illustrated that genetic clustering of TULV is largely according to geographic regions [22,33].

## 5. Conclusions

Here, we screened 621 small mammals for their orthohantavirus infection rate. Interestingly, we only identified the relatively benign TULV species, a finding that is contrary to the expectation risen by patients with episodes of haemorrhagic fever in Kazakhstan hospitals. Knowledge on the pathogenicity of TULV and the impact of TULV-associated disease in humans is limited. Only few cases, mostly mild, were described in Europe, some of them in immunocompromised patients [10,11,12,13,49]. In certain risk groups, e.g., forest workers, a higher antibody prevalence against TULV was found in comparison to the normal population [11,12]. However, the severe cases of HFRS observed in the hospitals in West Kazakhstan are most probably not induced by an infection with TULV but rather by PUUV. The exact endemic areas for this virus in Kazakhstan remain obscure.

## Figures and Tables

**Figure 1 viruses-14-01258-f001:**
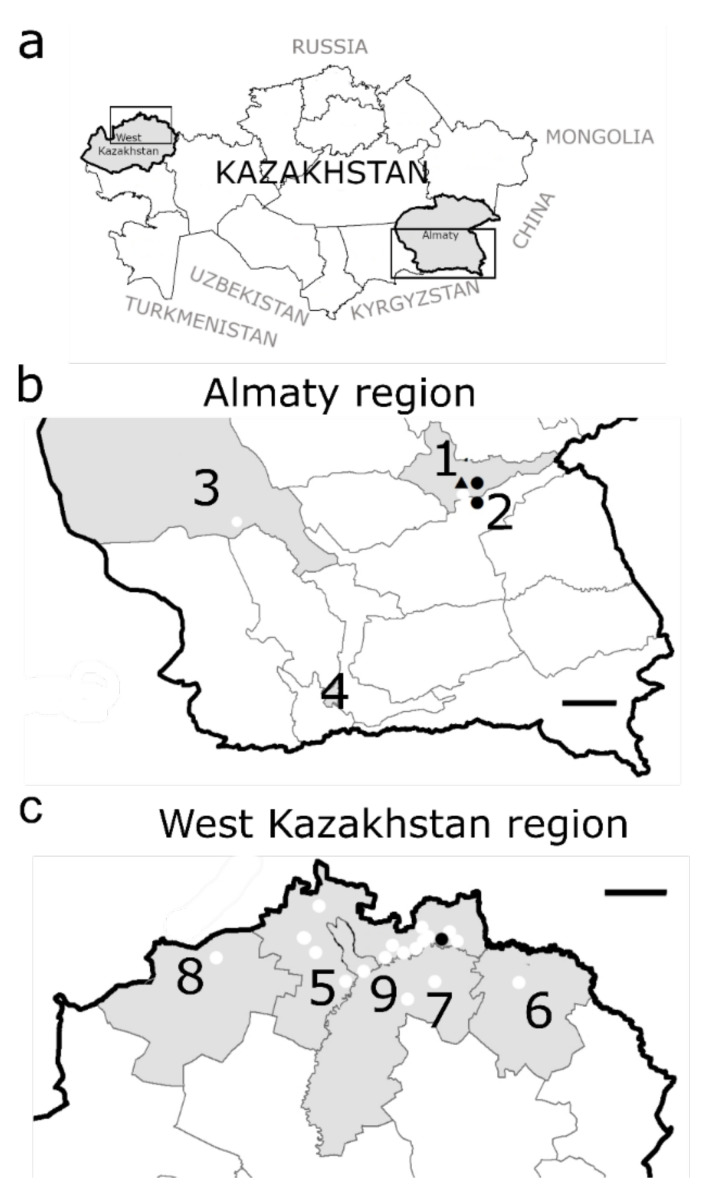
Geographical location of the sampling points for small mammals in Kazakhstan. (**a**): Kazakhstan is divided in 14 oblasts (=regions) and located in Central Asia. (**b**): Almaty region and Almaty city: **1.** Tekeli city: 2 trapping sites; **2.** village Rudniychniy: 1 trapping site; **3.** village Bakanas: 1 trapping site; **4.** Almaty city: 7 trapping sites; (**c**): West Kazakhstan region: **5.** district Bayterek: 12 trapping sites; **6.** district Borili: 1 trapping site; **7.** district Terekti: 2 trapping sites; **8.** district Taskala: 1 trapping site; **9.** Oral city: 3 trapping sites. Sampling locations: white dots. Species and location of infected rodents: ● *Microtus arvalis*, ▲ *Dryomys nitedula*. Black frames = regions magnified in (**b**) and (**c**), size marker = 150 km.

**Figure 2 viruses-14-01258-f002:**
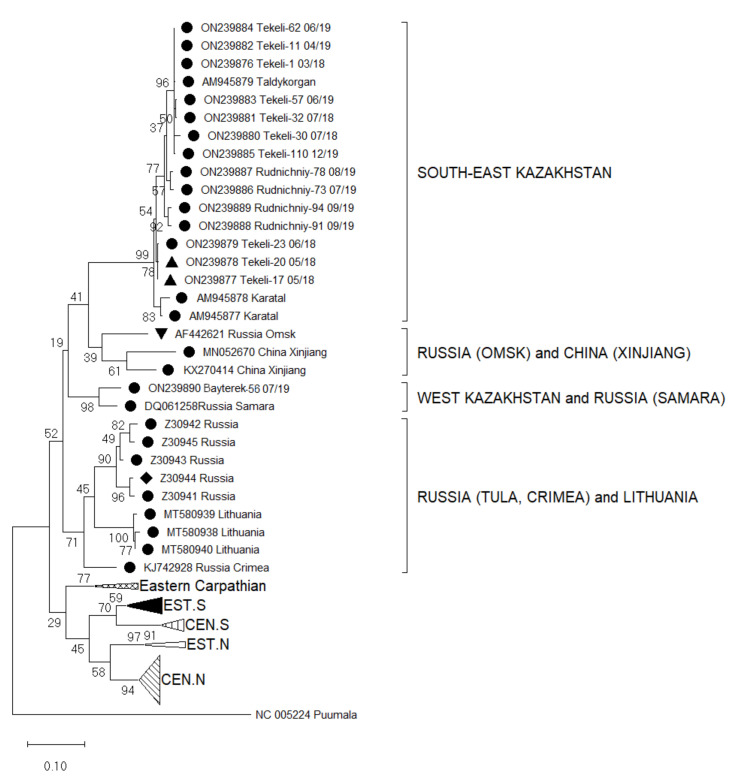
Phylogenetic analysis by Maximum Likelihood method of the S segments (346 nucleotides (nt), positions of sequences 715–1061 nt in regard to the reference sequence AM945879) of Tula virus in Kazakhstan. The tree with the highest log likelihood (−5756.38) is shown. The percentage of trees in which the associated taxa clustered together is shown next to the branches. This analysis involved 92 nucleotide sequences: Central North (CEN.N): KU139579, KU139576, KU139577, KU139578, DQ662094, HQ697346, HQ697344, HQ697347, HQ697351, GU300137, GU300136, EU439952, EU439947, EU439949, EU439948, EU439950, EU439946, EU439951, KU139534, KU139535, KU139537, KU139538, KU139598, KU139595, KU139596, KU139599, KU139529, KU139528, KU139531, KU139530, KU139533, DQ662087, DQ768143; Eastern North (EST.N): AF063897, AF289819, AF289820, AF289821; Central South (CEN.S): AF164093, HQ697350, HQ697348, HQ697349, HQ697355, HQ697353, HQ697354, HQ697357; Eastern South (EST.S): AJ223601, U95312, KF184327, KF184328, NC005227, Z69991, Z49915, Z48741, AJ223600, Z48574, KU139560; Eastern Carpathian: AF017659, Y13980, KF557547, Y13979; Russia Tula: Z30941, Z30942, Z30943, Z30944, Z30945; Russia Crimea: KJ742928; Lithuania: MT580938, MT580939, MT580940; Russia Samara: DQ061258; Russia Omsk: AF442621; China Xinjiang: MN052670, KX270414; South-East Kazakhstan: AM945877, AM945878, AM945879, outgroup Puumala NC005224. Host Species: ● *Microtus arvalis*, ▲ *Dryomys nitedula*, ♦ *Microtus rossiaemeridionalis*, ▼ *Microtus gregalis*.

**Figure 3 viruses-14-01258-f003:**
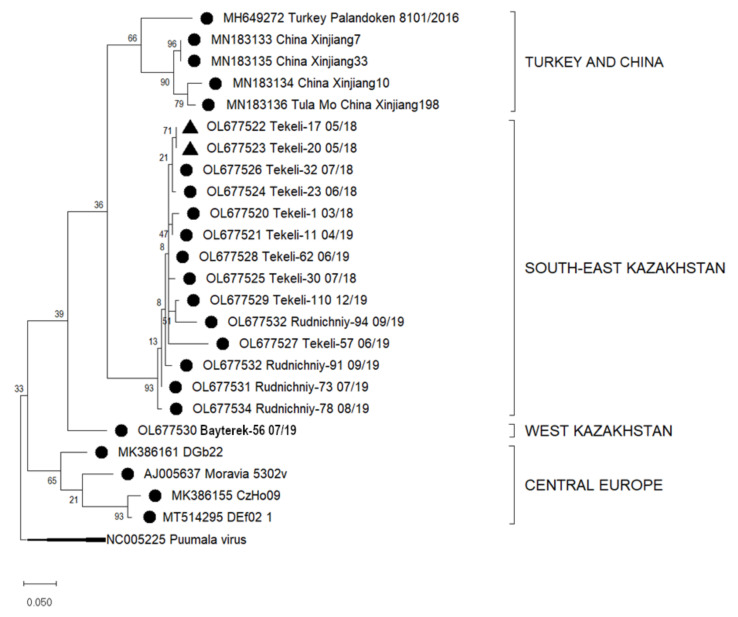
Phylogenetic analysis by Maximum Likelihood method of the L segments (184 nucleotides (nt), positions of sequences 5187–5371 nt in regard to the reference sequence NC005226) of Tula virus in Kazakhstan. The tree with the highest log likelihood (−1345.67) is shown. The percentage of trees in which the associated taxa clustered together is shown next to the branches. This analysis involved 25 nucleotide sequences: Turkey: MH649272; China: MN183133, MN183135, MN183134, MN183136; Europe: AJ005637, MK386161, MK386155, MT514295, outgroup Puumala NC005225. Host Species: ● *Microtus arvalis*, ▲ *Dryomys nitedula.*.

**Table 1 viruses-14-01258-t001:** All species captured in snap traps in the sampling areas of interest.

Small Mammal Species	West Kazakhstan (19 Trapping Sites)	Almaty Region (4 Trapping Sites)	Almaty City (7 Trapping Sites)
*Microtus arvalis* (Common vole)	13	72	1
*Myodes glareolus* (Bank vole)	12	0	0
*Microtus kirgisorum* (Tien Shan vole)	0	0	49
*Apodemus uralensis* (Ural or Pygmy field mouse)	128	84	47
*Mus musculus* (House mouse)	62	27	39
*Rattus norvegicus* (Brown rat)	0	0	39
*Meriones meridianus* (Midday jird)	0	2	0
*Dryomys nitedula* (Forest dormouse)	2	13	0
*Sorex araneus* (Common shrew)	1	0	0
*Sorex minutus* (Eurasian pygmy shrew)	0	1	1
*Crocidura suaveolens* (Lesser white-toothed shrew)	0	0	28
**Total**	**218**	**199**	**204**

**Table 2 viruses-14-01258-t002:** Result of the molecular biological screen for orthohantavirus RNA among small mammals captured in the regions of interest.

Small Mammal Species	Total Collected	Sex Ratio Male/Female	Number of Positive Samples (Male/Female)	Percentage of Positive Samples [%]
*Microtus arvalis*	86	40/46	13 (8/5)	15.1
*Dryomys nitedula*	15	7/8	2 (1/1)	13.3
*Myodes glareolus*	12	11/1	0	0
*Microtus kirgisorum*	49	26/23	0	0
*Apodemus uralensis*	259	163/96	0	0
*Mus musculus*	128	83/45	0	0
*Rattus norvegicus*	39	16/23	0	0
*Meriones meridianus*	2	2/0	0	0
*Sorex araneus*	1	0/1	0	0
*Sorex minutus*	2	1/1	0	0
*Crocidura suaveolens*	28	15/13	0	0
**Total**	**621**	**364/257**	**15 (9/6)**	**2.4**

**Table 3 viruses-14-01258-t003:** Nucleotide sequence identity of the partial Tula virus (TULV) S-segments detected from Kazakhstan in comparison with published sequences from other Eurasian regions (%).

S Segment Cluster	South-East Kazakhstan	China (Xinjiang)/ Russia (Siberia)	Russia (Tula and Crimea)	West Kazakhstan	Russia (Samara)
South-East Kazakhstan	94.3–100	78.9–99.4	78.9–99.4	78.9–99.4	75.8–99.1
China (Xinjiang)/ Russia (Siberia)		84.5–87.5	81.6–98.5	82.1–87.5	79.9–88.9
Russia (Tula and Crimea)			87.5–98.5	84.5–98.5	85.6–97.9
West Kazakhstan				100	93.4
Russia (Samara)					100

**Table 4 viruses-14-01258-t004:** Nucleotide sequence identity of the partial Tula virus (TULV) L segment sequences in Kazakhstan and other Eurasian regions (%).

L Segment Cluster	Turkey and China	South-East Kazakhstan	West Kazakhstan	Central Europe
Turkey and China	85.9–100	80–99.3	81.6–85.9	78.3–97.2
South-East Kazakhstan		89.3–100	80.6–99.3	76.9–88.3
West Kazakhstan			100	79.4–97.2
Central Europe				87–97.2

## Data Availability

The data used and/or analysed during the current study are available from the corresponding author on reasonable request. All determined sequences were uploaded to GenBank and are accessible as OL677520, OL677521, OL677522, OL677523, OL677524, OL677525, OL677526, OL677527, OL677528, OL677529, OL677530, OL677531, OL677532, OL677533, OL677534, ON239876, ON239877, ON239878, ON239879, ON239880, ON239881, ON239882, ON239883, ON239884, ON239885, ON239886, ON239887, ON239888, ON239889, ON239890, ON513437, ON513438, and ON513439.

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
