# Peer review of "Molecular Characterisation and Phylogeny of Tula Virus in Kazakhstan"

_viruses, 2022, doi:10.3390/v14061258_

Round 1

Reviewer 1 Report

I am satisfied with the revision of this manuscript and also with the answers to my concerns mentioned in my previous review. After the revision this manuscript improved significantly.

I have a small concern:

Line 175, 180-181: Why some of the guanine "g" nucleotide in the primer sequences are in Lower case? is their any specific reason?

Author Response

A big thank you to the reviewer for appreciating our improvements of the manuscript. The lower case "g" in the primer sequences indicate some wobble pairings with some orthohantaviruses. Since they are not of any consern to the viruses we detected we changed the lower case to an upper case G in the sequences.

Reviewer 2 Report

The authors revised the article, took into account all my previous comments and made all the proposed changes to the manuscript. I think that the article is ready for publication.

Author Response

Thank you

This manuscript is a resubmission of an earlier submission. The following is a list of the peer review reports and author responses from that submission.

Round 1

Reviewer 1 Report

General comments:

The manuscript ‘Molecular epidemiology of Tula virus in Kazakhstan’ is of interest because it is the first investigation of ortohantaviruses prevalence in this country.

Main comments:

In my opinion, the study was conducted at a fairly high level, the results look reliable and convincing. However, I have a number of questions and comments listed below.

1. Line 2 – According to the ICTV nomenclature , the virus is called ‘Tula orthohantavirus’.

2. Line 56 – better to use the term ‘negative polarity RNA’.

3. Fig. 1 has low quality.  The text and numbers on the figure need to be made larger and clearer.

4. Lines 129-130 – it is unclear from the text under what conditions (temperature and time) the captured animals were transported at and what measures were taken to preserve the biological material.

5. Lines 153 – it is not specified whether the received sequences are placed in the database (GenBank).

6. Lines 155-157 – it is not specified how many and which reference nucleotide sequences were included in the analysis. If the sequences were obtained from the GenBank, then it is necessary to specify their accession numbers. It is also not specified which sequences were used as outgroup.

7. Lines 163-166 – species names should be italic format.

8. Fig. 2 и Fig. 3 – Why different types of orthohantaviruses are used as outgroup for phylogenetic analysis of S and L segments? It is necessary to use one type of orthohantavirus as outgroup on all the infered trees.

9. Fig. 2 и Fig. 3 – The location of the sequenced fragment in the genome and the position numbers according to the reference sequence are not specified.

10. Better to use terms ‘clade’, ‘subclade’, ‘branch’ etc in place of ‘section’, ‘subheading’.

11. Throughout the text – there is no need to repeat the usual and Latin names of the mammal every time.

12. Line 208 – it is unclear whether the authors mean only the sequences they identified or all sequences including the references used in the phylogenetic analysis. Can they make any suggestion explaining the identity of amino acid sequences in different strains? For example, on Table 3 the authors give the values of nucleotide sequences identity varing from 75.8% to 99.4%, but aa sequences are 100% identical. Does this mean that all nucleotide substitution are synonymous?

13. Line 274 – extra comma after the word ‘TULV’. The next sentence should start with a capital letter.

Author Response

Reviewer 1:

General comments:

The manuscript ‘Molecular epidemiology of Tula virus in Kazakhstan’ is of interest because it is the first investigation of ortohantaviruses prevalence in this country.

Main comments:

In my opinion, the study was conducted at a fairly high level, the results look reliable and convincing. However, I have a number of questions and comments listed below.

  1. Line 2 – According to the ICTV nomenclature, the virus is called ‘Tula orthohantavirus’.

Our answer: Thank you for this comment. Indeed its correct in ICTV nomenclature its Tula orthohantavirus. However, there was an important publication two years ago, that states that TULA orthohantavirus includes Tula virus and Adler virus. Since we deliberately only want to report on Tula and not on Adler virus we decided to designate it Tula virus throughout the paper. Find this publication here:  Kuhn JH, Adkins S, Alioto D, et al. 2020 taxonomic update for phylum Negarnaviricota (Riboviria: Orthornavirae), including the large orders Bunyavirales and Mononegavirales. TABLE 7. Arch Virol. 2020 Dec;165(12):3023-3072. doi: 10.1007/s00705-020-04731-2. Epub 2020 Sep 4. PMID: 32888050; PMCID: PMC7606449.

  1. Line 56 – better to use the term ‘negative polarity RNA’.

Our answer: Thanks for pointing out this impreciseness. We adapted it accordingly.

  1. Fig. 1 has low quality.  The text and numbers on the figure need to be made larger and clearer.

Our answer: Thanks to the reviewer for pointing this out. Figure design was revisited and exported into a high-quality file (600dpi).

  1. Lines 129-130 – it is unclear from the text under what conditions (temperature and time) the captured animals were transported at and what measures were taken to preserve the biological material.

Our answer: Sorry for don’t being precise enough in this section. Our mouse capture procedure started in the evening with placing the traps with the baits. The next day, in the early morning (~5 a.m.) captured small mammals were placed separate plastic bags and collected in cooler boxes on dry ice. Mice were transferred as fast as possible to the laboratory and kept cold until dissected the same day.

  1. Lines 153 – it is not specified whether the received sequences are placed in the database (GenBank).

Our answer: Thanks for pointing this out. Of course, all sequences will be uploaded to GenBank, once the paper is accepted. We inserted a sentence at the end of the paper in the data availability statement to link to these sequences. The sequences are: OL677520, OL677521, OL677522, OL677523, OL677524, OL677525, OL677526, OL677527, OL677528, OL677529, OL677530, OL677531, OL677532, OL677533 and OL677534. They remain blocked in GenBank till the paper is finally accepted.

  1. Lines 155-157 – it is not specified how many and which reference nucleotide sequences were included in the analysis. If the sequences were obtained from the GenBank, then it is necessary to specify their accession numbers. It is also not specified which sequences were used as outgroup.

Our answer: Thanks for pointing out this lack of detailinformation. Nucleotide sequence alignments were generated with the CLUSTAL Omega program. Nucleotide sequences were aligned in Bioedit 7.2.5. Nucleotide sequence similarity searched in the public database National Center for Biotechnology (www.ncbi. nlm.nih.gov/blast/), were assessed by the Basic Local Alignment Search Tool, using BLASTn, and BLASTn optimized for highly similar sequences (MEGABLAST). This analyses involved 30 nucleotide sequences for S segment with accession numbers AM945879, AM945877, AM945878, MN052670, KX270414, AF442621, Z30945, Z30942, Z30943, Z30941, Z30944, KJ742928, DQ061258, Y13979, Y13980, AF017659, AF289820, AF289821, AF289819, AF063897, AF063892, MH649270, AF164093, AF164094, AJ223601, AJ223600, Z49915, Z48573, Z48741, Z48574 and 9 nucleotide sequences for L segment with accession numbers: MH649272, MN183133, MN183135, MN183134, MN183136, MK386161, AJ005637, MK386155, MT514295 and as a outgroup Puumala S and L segment complete genes were used NC005224, NC005225. We updated this information in the text.

  1. Lines 163-166 – species names should be italic format.

Our answer: Sorry for this flaw. We set everything according to Linnès’ regulations.

  1. Fig. 2 и Fig. 3 – Why different types of orthohantaviruses are used as outgroup??? for phylogenetic analysis of S and L segments? It is necessary to use one type of orthohantavirus as outgroup on all the infered trees.

Our answer: Thank you for pointing out this weakness in the data sources. We now exchanged the outgroup sequences in both figures and selected for S and L a complete genome of a Puumala S and L segment. Accession numbers NC005224, NC005225.

  1. Fig. 2 и Fig. 3 – The location of the sequenced fragment in the genome and the position numbers according to the reference sequence are not specified.

Our answer: Thanks for raising this issue. The segments we analysed were for the L segment 184 nt and for the S segment 346 nt. Those segment lengths were also used for various other biosurveillance studies.
Within the S segment they are spanning the region 634 – 1050 of the 1910 nt long reference sequence. Within the L segment they are spanning the region 5161 – 5389 of the approximately 6500 nt long reference sequence.

  1. Better to use terms ‘clade’, ‘subclade’, ‘branch’ etc in place of ‘section’, ‘subheading’.

Our answer: Thank you for pointing out this sloppiness in our wording. We revised the entire text and changed the terms accordingly and are now using clades und subclades as our preferred term.

  1. Throughout the text – there is no need to repeat the usual and Latin names of the mammal every time.

Our answer: Thank you for pointing out this potential improvement for readability. We now mention the common names once in the introduction-table and then only stick to the latin names throughout the rest of the paper.

  1. Line 208 – it is unclear whether the authors mean only the sequences they identified or all sequences including the references used in the phylogenetic analysis. Can they make any suggestion explaining the identity of amino acid sequences in different strains? For example, on Table 3 the authors give the values of nucleotide sequences identity varing from 75.8% to 99.4%, but aa sequences are 100% identical. Does this mean that all nucleotide substitution are synonymous?

Our answer: Indeed, the text was a little cryptic here. We compared all our sequences we collected in the regions and clustered to the according subclades to sequences published elsewhere in neigbouring regions.
Truly, interestingly, although we sometimes had only down to 75% sequence identity, the mutations were mostly in the third position of the triplet and hence caused silent mutations. That’s why the aa
sequence mostly remained unaffected, only in the isolates Tekeli-110 (1) 12/19 and Rudnichniy-94 (3) 09/19 show at position 1760 a P versus R mutation and at position 1773 a K versus E aberration.

  1. Line 274 – extra comma after the word ‘TULV’. The next sentence should start with a capital letter.

Our answer: Thanks for pointing out this grammar issue. We completely revised the section and hope it now meets grammatical standards.

Reviewer 2 Report

My main question concerns the primers and the results related to the use of them. This has to be clarified. For the S-segment RT-PCR the authors use DOBV-specific primers. For the L-segment they use pan-hanta primers. It is not easy to understand how they get TULV S-segment sequences from this.

Lines 131-132. Here it says that lung, heart, brain, kidney, liver, spleen, ears and transudate were collected, but the authors only analyze lung tissue. Why? Please give a reason for why lung tissue was used, and not any other organ. Would there be a potential for more positives if other organs were to be analyzed?

Lines 140-149. Why are the primers written with alternating upper- and lower case?

Lines 140-142. The DOBV M6 and M8 primers are DOBV-specific according to reference 27. Yet the aim of the study was (on lines 101-102) “to investigate small mammals for the presence of orthohantaviruses by molecular biological methods….” Why use DOBV-specific primers?

Line 169. “Out of all 621 collected small mammals 15 (2.4%) were positive for Orthohantavirus”. By what RT-PCR were they positive? I guess it was by using the L-segment RT-PCR? Because the S-segment PCR had DOBV-specific primers. But I don’t understand the flow of results and experiments. It should be better explained. Is it so that the positive samples were detected by the pan-hanta L1 RT-PCR only? But why use a DOBV specific RT-PCR for the S-segment? On lines 176-178 the authors claim to analyze sequences from S-segments from Tula virus, but on lines 150-151 they write that they sequence the RT-PCR products? Could the authors please explain how they can get Tula virus sequences by using DOBV-specific primers? And if so, add some results regarding this.

Line 176. How long were the sequences?

Line 234-    In the discussion there is nothing on the method they used for detection (RT-PCR). Were there any limitations? Can the pan-hanta used detect all hantaviruses? Could there be new variants in this region? Why did they not detect any PUUV? Was it the method or the sampling sites? Or the season?

In general, there are many typos and mistakes in grammar in the text – the language need improvement.

Author Response

Reviewer 2:

My main question concerns the primers and the results related to the use of them. This has to be clarified. For the S-segment RT-PCR the authors use DOBV-specific primers. For the L-segment they use pan-hanta primers. It is not easy to understand how they get TULV S-segment sequences from this.

Lines 131-132. Here it says that lung, heart, brain, kidney, liver, spleen, ears and transudate were collected, but the authors only analyze lung tissue. Why? Please give a reason for why lung tissue was used, and not any other organ. Would there be a potential for more positives if other organs were to be analyzed?

Our answer: Thank you for raising this issue. Indeed, we only analysed the lungs from our captured mice although we collected much more organs. Most of the studies reported existing orthohantaviruses in lung and kidney tissues and also some studies demonstrated also positive orthohantaviruses in liver, spleen, heart, brain tissues (Schmidt-Chanasit et al., doi:10.1128/JVI.01226-09, Madai et al., 2021 doi: 10.3390/v13040570, He et al., doi.org/10.3389/fvets.2021.748232). As far as it was a first study on molecular-biology investigation of orthohantaviruses in Kazakhstan we started our screening with lung tissues, otherwise the workload and the consumption of lab-plasticware (that is limited in these times, especially in Kazakhastan) would have exceeded our possibilities.  

Lines 140-149. Why are the primers written with alternating upper- and lower case?

Our answer: thanks for pointing out this impreciseness. The lower-case g’s were caused by the effort to improve of the readability while generating the primers and somehow this never got corrected. Now the sequences are all uniformly in upper case.

Lines 140-142. The DOBV M6 and M8 primers are DOBV-specific according to reference 27. Yet the aim of the study was (on lines 101-102) “to investigate small mammals for the presence of orthohantaviruses by molecular biological methods….” Why use DOBV-specific primers?

Our answer: Indeed, the reviewer is absolutely right, that the M6 and M8 primers were originally designed to detect DOBV. But as many published studies and broad analysis with many hantaviruses in our laboratory proved, DOBV-M6 & M8 detects because of its Wobble bases also other Orthohantaviruses (Data not shown). Therefore, DOBV M6-M8 primers were used to investigate small mammals for the presence of orthohantaviruses (Schmidt-Chanasit J, Essbauer S, Petraityte R, et al. Extensive host sharing of central European Tula virus. J Virol. 2010;84(1):459-474. doi:10.1128/JVI.01226-09). The actual Differentiation and Determation of gained Fragments is then performed by Sequencing and only so the genotype of the virus can be determined.

Line 169. “Out of all 621 collected small mammals 15 (2.4%) were positive for Orthohantavirus”. By what RT-PCR were they positive? I guess it was by using the L-segment RT-PCR? Because the S-segment PCR had DOBV-specific primers. But I don’t understand the flow of results and experiments. It should be better explained. Is it so that the positive samples were detected by the pan-hanta L1 RT-PCR only? But why use a DOBV specific RT-PCR for the S-segment? On lines 176-178 the authors claim to analyze sequences from S-segments from Tula virus, but on lines 150-151 they write that they sequence the RT-PCR products? Could the authors please explain how they can get Tula virus sequences by using DOBV-specific primers? And if so, add some results regarding this.

Our answer: The are sorry that we were unclear in our choice of wording here. As already clarified in the remark one question above the DOBV primers for the S-segment are pan hanta primers. So we screened all our samples with the s-Segment primers M6 and M8 after performing a reverse transcription step. Positive samples were identified on a gel. Amplified DNA from those PCR runs were then purified and sequenced. Only after sequencing the determination of the genotype was possible (Fig2). Furthermore, frompositive samples some isolate was again run on a PCR to amplify also the L segment, this time using a real time RT PCR – which might seem confusing to the reader (Figure 3).

Line 176. How long were the sequences?

Our answer: Thanks for raising this issue, which was also an issue among other reviewers. The segments we analysed were for the L segment 184 nt and for the S segment 346 nt. Those segment lengths were also used for various other biosurveillance studies.

Line 234- In the discussion there is nothing on the method they used for detection (RT-PCR). Were there any limitations? Can the pan-hanta used detect all hantaviruses? Could there be new variants in this region? Why did they not detect any PUUV? Was it the method or the sampling sites? Or the season?

Our answer: Thanks for raising this remark but we are convinced, that this wont remain an issue since we now revised the text in the methods and results section where we explain how our workflow was. However, to specify once more, the DOBV M6, M8 primers detect all orthohantaviruses as well as panHanta 1 a,b,c,d primers detect Andres, Puumala, Hantaan, Dobrava, Seoul, Tula, Altai viruses. There weren’t any limitations as we use both PCR for detect S and L segments.

In general, there are many typos and mistakes in grammar in the text – the language need improvement.

Our answer: Thank you for raising this issue. We revised the entire text and tried to eradicate all typographic and grammatical mistakes. We hope it meets now the English grammar minimal standards.

Reviewer 3 Report

The manuscript “Molecular epidemiology of Tula virus in Kazakhstan” by Tukhanova et al. describe the molecular characterization of the circulating TULV in geographically important regions of the Kazakhstan. Authors also identified that the TULV were clustered into two different phylogenetic clades and also identified Dryomys nitedula as a novel host.

There are few issues that need to be addressed:

First, English language and spellings has to be checked and corrected/improved throughout the manuscript. Please have someone fluent in English proofread this manuscript thoroughly. 

Line 230-233: Please, remove the personal notes,  (This section.......can be drawn.) before submitting your manuscript for publication.

Authors didn't mention whether they have submitted the sequences of the TULV genes used in this study to any public repository or not.

After Identifying positive samples, authors should have amplified and sequenced the full length genes and used it for the phylogenetic analysis and amino acid alignment for better/clear genetic picture of the circulating TULV strains in these regions.

Authors should mention, what are the parameters they have used for the identifications of the rodent species? it is important to mention as authors are claiming to identify a novel host of TULV in the region.

Authors should have shown the amino acid alignment in support of their results.

Author Response

Review 3:

The manuscript “Molecular epidemiology of Tula virus in Kazakhstan” by Tukhanova et al. describe the molecular characterization of the circulating TULV in geographically important regions of the Kazakhstan. Authors also identified that the TULV were clustered into two different phylogenetic clades and also identified Dryomys nitedula as a novel host.

There are few issues that need to be addressed:

First, English language and spellings has to be checked and corrected/improved throughout the manuscript. Please have someone fluent in English proofread this manuscript thoroughly.

Our answer: Thank you for raising this issue. We revised the entire text and tried to eradicate all typographic and grammatical mistakes. We hope it meets now the English grammar minimal standards.

Line 230-233: Please, remove the personal notes,(This section.......can be drawn.) before submitting your manuscript for publication.

Our answer: Thank you for raising this mishap. We revised the entire piece and hope that we now found all wrong editings.

Authors didn't mention whether they have submitted the sequences of the TULV genes used in this study to any public repository or not.

Our answer: Thanks for pointing this out. Of course, all sequences will be uploaded to GenBank, once the paper is accepted. We inserted a sentence at the end of the paper in the data availability statement to link to these sequences. The sequences are: OL677520, OL677521, OL677522, OL677523, OL677524, OL677525, OL677526, OL677527, OL677528, OL677529, OL677530, OL677531, OL677532, OL677533 and OL677534. They remain blocked in GenBank till the paper is finally accepted.

After Identifying positive samples, authors should have amplified and sequenced the full length genes and used it for the phylogenetic analysis and amino acid alignment for better/clear genetic picture of the circulating TULV strains in these regions.

Our answer: Thank you for this remark. Indeed, our initial plan was to obtain entire sequences but this was limited by several reasons: Due to existing trade issues it exhaustive to import new primer sequences to Kazakhstan and production of primers in the countries is sometimes of limited success due to varying production quality. Sencond, and most important to our research question, almost all of the sequences in the neigbouring states that are published are also only such short fragments as we had used. Hence comparing putative complete sequences to the stumps available online would have in the end brought the same insight on the local variation of the virus. However, we are currently applying for funding to expand the biosurveillance study to entire Central Asia and there it will be definitely our intention to obtain entire sequences.

Authors should mention, what are the parameters they have used for the identifications of the rodent species? it is important to mention as authors are claiming to identify a novel host of TULV in the region.

Our answer: Thank you for pointing out this important question. Indeed on the determination of the rodent species a special attention was put. The identification was performed by a specialist for endemic rodents in the regions, employed by the Kazakhstan Antiplage station-network. Identification of rodent species include external signs of rodents: size, tail, forelimbs, hind legs, the fingers, hairline and internal the structure of skull and teeth parameters as described Gromov M, Erbajeva M. The Mammals of the Russia and adjacent territories 1995 (Lagomorphs and Rodents). After the finding of the new host species, we revisited our notes on the necropsy and the expert claimed, that his determination on that given animal was without any doubt.

Authors should have shown the amino acid alignment in support of their results.

Our answer: Thanks for this comment. Indeed, we planned to insert the aa alignments into the supplement. But since only in two isolates there were aberrations in the sequence the whole figure contained little extra information. That’s why we desided to leave this figure out for the sake of the elegance of the paper. However, if the reviewer insists on this figure, we are happy to incorporate it.

Round 2

Reviewer 3 Report

The authors have upgraded the manuscript significantly.